# Development and In Vivo Assessment of 4-Phenyltellanyl-7-chloroquinoline-loaded Polymeric Nanocapsules in Alzheimer’s Disease Models

**DOI:** 10.3390/brainsci13070999

**Published:** 2023-06-28

**Authors:** Ana Cláudia Funguetto-Ribeiro, Kelly Ayumi Nakama, Mikaela Peglow Pinz, Renata Leivas de Oliveira, Manoela do Sacramento, Flávia S. Oliveira Pereira, Simone Pinton, Ethel Antunes Wilhelm, Cristiane Luchese, Diego Alves, Daiana Silva Ávila, Sandra Elisa Haas

**Affiliations:** 1Biochemistry Graduate Program, Federal University of Pampa—UNIPAMPA, Uruguaiana 97501-970, Brazil; acfunguetto@gmail.com (A.C.F.-R.); flaviapereira.aluno@unipampa.edu.br (F.S.O.P.); simonepinton@unipampa.edu.br (S.P.); daianaavila@unipampa.edu.br (D.S.Á.); 2Pharmaceutical Science Graduate Program, Federal University of Pampa—UNIPAMPA, Uruguaiana 97501-970, Brazil; kelly.ayumi@hotmail.com; 3Biochemistry and Bioprospecting Graduate Program, Biochemical Pharmacology Research Laboratory (LaFarBio), Neurobiotechnology Research Group (GPN), Chemical, Pharmaceutical and Food Science Center (CCQFA), Federal University of Pelotas—UFPel, Pelotas 96010-900, Brazil; mikaelappinz@gmail.com (M.P.P.); renataleivas15@hotmail.com (R.L.d.O.); ethelwilhelm@yahoo.com.br (E.A.W.); cristiane_luchese@yahoo.com.br (C.L.); 4Clean Organic Synthesis Laboratory (LASOL), Center for Chemical, Pharmaceutical and Food Sciences (CCQFA), Federal University of Pelotas—UFPel, Pelotas 96010-900, Brazil; manoelasacramento@hotmail.com (M.d.S.); dsalves@gmail.com (D.A.)

**Keywords:** neurodegeneration, neuroprotection, nanoparticles, organochalcogenium, tellurium

## Abstract

Alzheimer’s disease (AD) is the most common form of dementia in older people, and available treatments are palliative and produce undesirable side effects. The 4-phenyltellanyl-7-chloroquinoline (TQ) is an organochalcogen compound studied due to its pharmacological properties, particularly its antioxidant potential. However, TQ possesses some drawbacks such as low aqueous solubility and high toxicity, thus warranting the search for tools that improve the safety and effectiveness of new compounds. Here, we developed and investigated the biological effects of TQ-loaded polymeric nanocapsules (NCTQ) in an AD model in transgenic *Caenorhabditis elegans* expressing human Aβ_1–42_ in their body–wall muscles and Swiss mice injected with Aβ_25–35_. The NCTQ displayed good physicochemical properties, including nanometer size and maximum encapsulation capacity. The treatment showed low toxicity, reduced Aβ peptide-induced paralysis, and activated an endoplasmic reticulum chaperone in the *C. elegans* model. The Aβ injection in mice caused memory impairment, which NCTQ mitigated by improving working, long-term, and aversive memory. Additionally, no changes in biochemical markers were evidenced in mice, demonstrating that there was no hepatotoxicity in the tested doses. Altogether, these findings provide insights into the neuroprotective effects of TQ and indicate that NCTQ is a promising candidate for AD treatment.

## 1. Introduction

Alzheimer’s disease (AD) is the most common form of dementia in the elderly population. The main hallmarks of AD are extracellular amyloid-β peptide (Aβ) aggregation and neurofibrillary tangles formation [1]. Episodic memory impairment is one of the main aspects of this disease, and other symptoms include delayed recall and impaired language skills. Therapeutic alternatives attempting to counteract the damage caused by AD have yet to achieve permanent successful results, which include acetylcholinesterase inhibitors (AChE) (tacrine, donepezil, rivastigmine, and galantamine) and N-methyl-D-aspartate receptors (NMDARs) (memantine). However, these treatments produce undesirable side effects, thereby hindering patient adherence to treatment. Consequently, the search for new forms of medication is currently an urgent need.

Organochalcogen compounds comprise organic molecules that contain selenium, tellurium, sulfur, or oxygen in their structure. These compounds have gained notoriety over the years due to their pharmacological properties against neurodegenerative diseases, mainly in reversing and/or protecting memory impairment in animal models of AD [2,3,4]. Nevertheless, the utilization of organochalcogens in biological applications presents certain challenges, including concerns about their toxicity and specific physicochemical properties (such as low hydrophilicity, which affects its bioavailability). Recent studies have provided evidence that incorporating organoalcogen compounds into nanoparticles significantly enhances their selectivity and pharmacological effectiveness [5,6,7,8].

Our research group has been studying the pharmacological, toxicological, and molecular effects of organochalcogen compounds in *Caenorhabditis elegans* and rodents [2,9,10,11,12,13], and tellurium organic compounds have demonstrated neuroprotective properties against Mn-induced neurotoxicity and in the AD model in mice [12,14,15]. However, it is important to note that pharmacologic studies performed with organotellurium compounds are relatively scarce, since their pharmacological window is narrow [16,17]. As a result, the development of new organotellurium compounds still needs to be explored. In this sense, we highlight the TQ (Figure 1), which showed promising pharmacological properties, since TQ modulated transcription factors SKN-1 (homologous to NRF2 in humans) and DAF-16 (homologous to FOXO3a in humans) and its downstream effectors, including the antioxidant enzymes SOD-3 and GCS-1 [13], which are both pathways required for the beneficial effects exerted by other neuroprotective therapies in transgenic *C. elegans* [18,19] and mammals [19,20].

Given the above, we hypothesized that nanoencapsulation of TQ could reduce toxicity and optimize its effectiveness by modulating physicochemical and biopharmaceutical properties or minimizing adverse effects and maximizing therapeutic efficacy. Polymeric nanocapsules have been shown to improve drug solubility, increase their bioavailability, and target specific tissues or cells, thereby reducing systemic toxicity and improving therapeutic efficacy [5,21,22]. 

The use of nanoencapsulated drugs is an opportune strategy for brain delivery. Different nanoparticles are described in the literature, including nanosystems such as nanotubes, nanospheres, liposomes, and nanocapsules. The use of polymeric nanocapsules has been reported as a reproducible method with good stability and biocompatibility [21,23,24,25]. Studies have demonstrated that polysorbate 80 (nonionic surfactant) causes a temporary and partial disruption of the blood–brain barrier, decreasing P-glycoprotein (P-gp) efflux activity and apolipoprotein-E (ApoE) levels which facilitate endocytosis in the brain [24,26], which may lead to substantial concentrations of these drugs in the central nervous system. In addition, studies have demonstrated an improved response in vivo using nanoencapsulated drugs targeting the brain, such as curcumin and meloxicam [27,28,29]. Thus, given that nanoencapsulated TQ may be a worthwhile strategy in AD drug development, this study aimed to develop and characterize the physicochemical properties of polymeric nanocapsules containing TQ (NCTQ) and test the potential neuroprotective effect of TQ and NCTQ in different models of AD.

## 2. Materials and Methods

### 2.1. Chemical and Reagents

TQ was obtained from Laboratório de Síntese Orgânica Limpa (LASOL, Pelotas, RS, Brazil) [30]. Acetonitrile and methanol were obtained from Tedia (Fairfield, CT, USA). The purified water was prepared using a Milli-Q Plus system (Millipore, Bedford, MA, USA). The Aβ_25–35_ was obtained from Sigma (St. Louis, MO, USA) and dissolved in sterile filtered water. All other reagents and chemicals were of pharmaceutical or special analytical grade.

### 2.2. Nanocapsules Preparation

The NCTQ and nanocapsules without TQ (NCBR) were obtained by interfacial deposition of poly(ε-caprolactone) (PCL) [21]. The organic phase was composed of PCL polymer, Span 60^®^ surfactant, TQ (1 mg/mL), caprylic/capric triglyceride, acetone, and ethanol and maintained at 35 ± 1 °C under magnetic stirring to solubilize all compounds. Afterward, this phase was injected into the aqueous phase containing distilled water and polysorbate 80 surfactant to produce the nanocapsules. 

### 2.3. Physicochemical Characterization of the Formulations

#### 2.3.1. Particle Size, Zeta Potential, and pH

The mean diameter (D_[4,3]_) and particle size distribution (span) of nanocapsules suspension were evaluated by laser diffraction analysis (Mastersizer 2000^®^, Malvern Instruments, Worcestershire, UK) (*n* = 3). The superficial charge was analyzed as a zeta potential by electrophoretic migration (NanoBrook 90Plus, BrookHaven^®^) (*n* = 3) from a 1:1000 dilution of samples in a pre-filtered NaCl solution (1 mM). The pH of the nanoformulations was evaluated immediately after preparation using a previously calibrated potentiometer (Hanna Instruments, São Paulo, Brazil) (*n* = 3).

#### 2.3.2. Drug Content Determination and Encapsulation Efficiency

Sample solutions containing NCTQ were prepared at the theoretical concentration of 10 µg/mL in a volumetric flask and maintained in an ultrasonic bath for 30 min to break the nanocapsules. The samples were filtered (0.45 µm, Millipore) before HPLC-PDA quantification (a method previously developed and validated according to Section 2.3.4). Encapsulation efficiency (EE%) was assessed by the ultrafiltration centrifugation method (Ultrafree, Millipore). The EE% was determined by the difference between the total concentration of each drug and its concentration in the aqueous phase using Equation (1) and given as a percentage of recovery (%) relating to the theoretical concentration.
(1)$$EE\%=\left(\frac{TQ_{total}-TQ_{free}}{TQ_{total}}\right)*100$$
where $TQ_{total}$ is the theoretical concentration of *TQ* and $TQ_{free}$ is the concentration of free *TQ* in the ultrafiltrate.

#### 2.3.3. HPLC-PDA Apparatus and Chromatographic Conditions

A Shimadzu^®^ High-Performance Liquid Chromatography (HPLC) system (Kyoto, Japan) equipped with a photodiode array (PDA) detector was used for separation and absorbance analysis, respectively. Data acquisition and system control were performed by analytical software (LC Solution, Release 1.22 SP1). The mobile phase comprised acetonitrile:water:triethylamine (90:10:0.3 *v*/*v*/*v*). The aqueous phase was adjusted to pH 7.0 with phosphoric acid. The TQ was separated using a chromatographic column at 25 ± 1 °C (5 μm, 4.6 × 150 mm; Nano Separation Technologies RP-18). The flow rate was defined as 1.0 mL/min, and TQ was detected at 325 nm after injecting 20 μL.

#### 2.3.4. Method Validation and Sample Preparation

The proposed method was validated based on the determination of specificity, linearity, precision, accuracy, robustness, and system suitability, according to general guidelines [31,32,33].

A standard TQ solution (400 μg/mL) was diluted with acetonitrile to obtain higher concentration solutions (5, 10, 15, 20, 30, 40, and 50 μg/mL). These flasks were kept in an ultrasonic bath for 10 min and filtered before injection (0.45 µm, Millipore^®^).

##### Linearity and Sensitivity

Linearity was analyzed by preparing three calibration curves of TQ solution in three different assays (5, 10, 15, 20, 30, 40, and 50 μg/mL) (*n* = 9). The correlation coefficient (R^2^) obtained by linear regression was evaluated to verify the suitability of the method. The calibration line was used to assess the limit of detection (LOD) and limit of quantitation (LOQ) [31,32,33].

##### Precision

Precision was determined by processing six independent TQ samples (10 μg/mL) on the same day (intra-day precision) or three different days (inter-day precision). The results were expressed as relative standard deviation percentage (RSD%).

##### Accuracy

Accuracy was determined by adding a single TQ concentration to 100 μL of NCBR in acetone, corresponding to 10 μg/mL (*n* = 3). The results were evaluated according to the recovery percentage of each sample.

##### Robustness

Robustness was determined by minimally changing some parameters of the analytical method, such as flow change (0.9 and 1.1 mL/min) and pH (6.8 and 7.2) (*n* = 3). These data were then analyzed according to values of Rt (retention time), T (tailing factor ≤ 2.0), k (retention factor ≥ 2.0), and N (theoretical plate number ≥ 2000) [31,32].

##### Specificity

Specificity was evaluated from samples containing NCBR to investigate possible interference of the excipients present in the nanoformulation. For this, 100 μL was added to a volumetric flask and diluted in acetonitrile to obtain 10 mL. The samples were maintained in an ultrasonic bath for 30 min and filtered before HPLC injection.

### 2.4. Alzheimer’s Disease Model in C. elegans

#### 2.4.1. *C. elegans* Maintenance and Treatment

This study used strains Bristol N2 (wild-type), SJ4005 (*hsp-4*::GFP), SJ4100 (*hsp-6p*::GFP + lin-15(+)), and CL2006 (pCL12(*unc-54*/human Aβ_1–42_) + pRF4), worms genetically modified for human Aβ expression. These animals were obtained from the *Caenorhabditis* Genetics Center (CGC, Minnesota, USA). Worms were maintained in nematode growth medium (NGM) plates seeded with *Escherichia coli* OP50 at 15 or 20 °C. After reaching the adult stage, pregnant hermaphrodites were washed using a bleaching solution (2.4% NaOH and 1% NaClO in distilled H_2_O) to obtain eggs. At the first larval stage (L1), 1500 worms were acutely exposed to TQ, NCTQ, or NCBR at 1 and 10 µM for 30 min. The control group was exposed to dimethyl sulfoxide (DMSO) at 5% (TQ vehicle). Afterwards, the worms were washed three times with M9 buffer (KH_2_PO4 0.02 M, Na_2_HPO_4_ 0.04 M, and NaCl 0.085 M) to remove the treatment. The animals were then placed on NGM plates seeded with *E. coli* OP50 until they reached the L4 larval stage (48 h after L1 stage). In the strain CL2006, the worms were exposed to an up-shift in temperature to a non-permissive 20 °C after the treatment to express the transgenic Aβ peptide in the body wall muscles (Figure 2).

#### 2.4.2. Toxicity Endpoints in *C. elegans*

Nanoformulation safety was evaluated by survival assay, counting the number of live worms in the plates after 48 h of treatment. Animal development was analyzed by obtaining images from five worms of each group and measuring the body length using the software ImageJ. One animal (*n* = 3) was transferred to new NGM plates with *E. coli* OP50 to determine brood size during 4 days of reproductive period, which is a parameter of reproductive toxicity and development. The results were normalized and expressed as a percentage of control. All assays were performed in duplicates, except for brood size (triplicates). All assays were repeated in three independent experiments.

#### 2.4.3. Chaperones HSP-4 and HSP-6 Expression

The strains SJ4005 and SJ4100 present a GFP reporter transgene controlled through hsp-4 and hsp-6 promoter, and the expression of these chaperones is localized in the endoplasmic reticulum and mitochondria, respectively. When the worms reached the larval stage L4, they were exposed to a temperature up-shift at 37 °C (heat shock) for 4 h to induce chaperone expression [34]. After this period, the worms were collected and washed with distilled water, then worms were anesthetized with levamisole (1 mM), placed on slides, and the images were obtained using fluorescent microscopy (Floid Cell Imaging Station^®^, Thermo Fisher Scientific, Waltham, MA, USA). The intensity fluorescence of 5 worms was quantified using the ImageJ software. This assay was conducted in triplicates and repeated three times. In addition, this method was performed in accordance with previously published studies by our research group [35].

#### 2.4.4. Aβ Peptide Aggregation Model in *C. elegans*

The strain CL2006 expresses the Aβ peptide in muscle cells, causing a locomotor impairment mentioned, (i.e., paralysis). Therefore, to verify the effects of the nanoformulations in an AD model, we performed a paralysis assay that measured worms’ response to mechanical stimuli. If the worms moved from the point of origin after stimuli (touch with a brush), they were not considered paralyzed, and if the animals only moved the head or pharynx, they were considered paralyzed. All assays were performed with 25 animals in duplicates and three independent experiments [35,36].

### 2.5. Alzheimer’s Disease Model in Mice

#### 2.5.1. Animals

Three-month-old male Swiss albino mice weighing 25–35 g were acquired from the Federal University of Pelotas (Brazil). We chose Swiss mice based on studies that demonstrated the effect of compounds in animal models of AD in this strain [9,26]. The animals were maintained at a constant temperature (22 ± 1 °C) and in a 12 h dark/light cycle and provided with food and water ad libitum. Animal care and experimental procedures were conducted in compliance with the Guide for the Care and Use of Laboratory Animals (NIH publication no. 8023) [37] and approved by the Committee on Care and Use of Animal Resources of the Federal University of Pelotas, Brazil (CEEA: 7046/2016). The number of animals and intensity of noxious stimuli used were the minimum needed to demonstrate the consistent effects of the treatments.

#### 2.5.2. Experimental Protocol

Mice were randomly divided into six experimental groups (6–8 animals/group): Sham, TQ, NCTQ, Aβ, Aβ + TQ, and Aβ + NCTQ. Thirty minutes before Aβ_25–35_ exposure, mice from the sham and Aβ groups received the NCBR dose (10 mL/kg), mice from the TQ and Aβ + TQ groups received the free TQ dose (1 mg/kg), and mice from the NCTQ and Aβ + NCTQ groups received the NCTQ dose at the same dosage. All treatments were intragastrically administered (i.g.) via oral gavage. After treatments, mice belonging to the Aβ, Aβ + TQ, and Aβ + NCTQ groups received Aβ (3 nmol/3 μL/per site by intracerebroventricular injection; i.c.v) [26], while the sham, TQ, and NCTQ groups received saline (3 μL/per site; i.c.v). The i.c.v infusion of Aβ or vehicle (saline) was administered according to a previous study [38]. Mice were treated with TQ, NCTQ, or NCBR every day until the 14th day. On the 15th day, mice were anesthetized by isoflurane inhalation for blood collection by a cardiac puncture, which was removed for ex vivo experiments. The experimental protocol is demonstrated in Figure 3.

#### 2.5.3. Behavioral Tests

##### Y-Maze Task

The Y-maze task was performed as described by Sarter et al. (1988) [39] and was used as a measure of spatial and working memory (8th day; Figure 3). The Y-maze apparatus consisted of a three-arm horizontal maze measuring 40 cm in length, 3 cm in width, and with walls 12 cm high. The three arms were positioned at 120° angles to each other, radiating out from a central point. The Y-maze test was performed on the eighth day of the experimental protocol (Figure 3). Mice were initially placed within one arm (A), and the sequence of arm entries (i.e., ABCCAB, where letters indicate arm codes) and the number of arm entries were manually recorded for each mouse over an 8-min period. Alternation was determined by observing successive entries into the three arms on overlapping triplet sets, where three different arms were entered. An actual alternation was defined as consecutive entries into all three arms (i.e., ABC, CAB, or BCA, but not BAB). An entry was counted when all four paws were placed within the boundaries of the arm. The percentage of alternation was calculated as follows: % Alternation = [(Number of alternations × 3)/(Total arm entries − 2)] × 100.

##### Object Recognition Task

To assess long-term memory (LTM), the object recognition task was used (12th day; Figure 3) [40]. Each mouse was submitted to a habituation session, and the LTM was performed 24 h after training (13th day; Figure 3), where mice explored a familiar object (A1) and a new object (B) for 5 min, and the total time spent exploring each object was determined. Cognitive performance was analyzed by calculating the exploratory preference, and data were expressed as a percentage as training = (A2/(A1 + A2)) × 100; LTM = (C/(A1 + C)) × 100.

##### Step-Down Inhibitory Avoidance

A step-down inhibitory avoidance task evaluated non-spatial and aversive LTM (14th day; Figure 3) [41]. In this one-trial learning task, the animals were put on a platform and received an electric shock (0.5 mA) for 2 s after stepping off the platform onto the grid. When the animals were tested 24 h later, they were exposed to the training apparatus, although no shock was delivered, and the transfer latency time was measured. The maximum transfer latency time was 300 s.

### 2.6. Ex-Vivo Assays

Mice were anesthetized with isoflurane, and blood samples were collected from the heart ventricle to obtain plasma and determine aspartate (AST) and alanine (ALT) aminotransferases (15th day; Figure 3). Plasma was obtained by centrifugation (900× *g*, 15 min). AST and ALT activities were determined in the plasma using a commercial kit (Bioclin, São Paulo, SP, Brazil).

### 2.7. Statistical Analysis

The *C. elegans* results were analyzed using one-way analysis of variance (ANOVA) and Tukey’s post hoc test for survival, body length, brood size, and chaperone expression. The paralysis rate was assessed by repeated measures ANOVA and Dunnett’s post hoc test (GraphPad Prism 7.04, GraphPad, San Diego, CA, USA). Mice data were analyzed by the GraphPad Prism 5 software, and data normality was evaluated by the D’Agostino and Pearson omnibus normality test. Statistical analysis was performed using one-way ANOVA followed by Tukey’s post hoc test. The data are expressed as mean ± standard error of the mean (SEM), and *p* < 0.05 values were considered statistically significant.

## 3. Results

### 3.1. Physicochemical Characterization of Nanocapsules

Both NCTQ and NCBR presented a white opalescent aspect commonly observed in nanosuspensions, known as the Tyndall effect. Laser diffraction analysis demonstrated particles on a nanometric scale with a small standard deviation in the diameter of their nanoparticles and span < 2.00. The NCTQ showed a small variation in D_[4,3]_ (240 ± 6 nm) regarding NCBR (231 ± 6 nm; Table 1). Both NCTQ and NCBR presented a negative zeta potential (−25.37 ± 1.52 and −24.36 ± 5.54, respectively; Table 1). Additionally, the pH evaluation of both formulations was approximately 6.00 (Table 1). The values obtained for EE% and drug content were close to the theoretical concentration of NCTQ formulation (1 mg/mL; Table 1).

### 3.2. Validation of the Analytical Methodology

The aqueous phase was composed of water containing 0.03% of triethylamine (*v*/*v*) (pH 6.2 adjusted with phosphoric acid) and the organic phase was acetonitrile (85:15 *v*/*v*). The flow rate was established as 1.0 mL/min with a total run time of 10 min. The chromatograms obtained showed a sharp and symmetric peak with good resolutions in these conditions. The maximum absorption of TQ was observed at 325 nm, as evidenced by the increase in the peak area at 8.20 min.

Linearity showed a linear relationship with the increase in concentration during drug quantification. The results demonstrated a linear relationship (R^2^ = 0.9996) with a representative linear equation established as y = 16,501x − 18,105. The precision was demonstrated as repeatability (intra-day) and intermediate precision (inter-day) (Table 2). The results showed that our method was found to be repeatable, according to guidelines (RSD < 2.0%) [31,32].

The robustness assay demonstrated minimal variations in the method caused expressive changes in the analyte chromatogram. The data show that slight pH and flow rate variations did not cause significant chromatographic changes to the proposed method, thus demonstrating its robustness. These values follow the guidelines and are summarized in Table 3 [31,32].

### 3.3. C. elegans Assays

#### 3.3.1. NCTQ Did Not Elicit Toxicity in *C. elegans*

The results obtained for survival rate, body length, and the number of larvae are shown below (Figure 4A–C). The results revealed that none of the acute treatments reduced worm survival rate at 1 or 10 µM in both free and nanoencapsulated forms of TQ (Figure 4A). In addition, no signs of toxicity in development and reproduction were observed (Figure 4B,C).

#### 3.3.2. TQ and NCTQ Increased Reticular Chaperone (HSP-4) Expression

Corroborating the paralysis results, TQ (1 and 10 µM) and the NCTQ (10 µM) increased GFP intensity of the reticular chaperone HSP-4 (Figure 5A–C), with NCBR (1 µM) also demonstrated this effect. However, when we evaluated GFP intensity in the strain SJ4100, which presents GFP-tagged HSP-6 and is localized in the mitochondria, we did not find any significant results in the modulation of this heat shock protein (Figure 5B–D).

#### 3.3.3. Treatment Reduced Aβ Peptide-Induced Paralysis Rate

The strain CL2006 was bioengineered with a constitutive muscle-specific promoter, accumulating β-immunoreactive deposits and intracellular amyloid (centrally involved in the AD pathogenesis), resulting in a progressive paralysis phenotype [42]. When we evaluated the locomotor dysfunction in the worms, we observed that both TQ and NCTQ reduced the paralysis rates by 10 μM (Figure 4D,E).

### 3.4. Mice Assays

#### 3.4.1. NCTQ Ameliorates Memory Impairment in Mice

##### Number of Arm Entries and Spontaneous Alternation Behavior

The effects of treatments on the behavioral parameters, specifically the number of arm entries and spontaneous alternation behavior in the Y-maze task of mice, are depicted in Figure 6A,B. The results demonstrated that the treatments did not significantly impact the number of arm entries (Figure 6A), suggesting that the treatment did not induce changes in motor function among the mice. Furthermore, the findings indicated that Aβ reduced spontaneous alternation behavior by approximately 25% compared to the sham group (Figure 6B). Another noteworthy finding was that the NCTQ treatment effectively prevented the reduction of Aβ-induced alternations in the Y-maze task, whereas TQ did not exhibit any effects (Figure 6B). Mice solely pretreated with TQ did not display any discernible differences in spontaneous alternation behavior during the Y-maze task (Figure 6B). However, the treatment with NCTQ alone significantly increased the number of spontaneous alternations (Figure 6B). These findings strongly suggest that NCTQ actively mitigates spatial and working memory impairments induced by Aβ.

###### Object Recognition Task

The effects of the treatments on the exploratory preference using the new object in the object recognition task (ORT) are illustrated in Figure 7A,B. There was no difference in the exploratory preference of objects among groups in the training phase (Figure 7A).

In the probe test, mice injected with Aβ showed reduced exploratory preference for the new object compared to the sham group (Figure 7B). Both TQ and NCTQ prevented the reduction of the preference exploratory for LTM (Figure 7B). These findings imply that TQ and NCTQ act in spatial and LTM impairment caused by Aβ.

###### Step-Down Inhibitory Avoidance Task

Figure 8A,B present the effect of treatments in the step-down inhibitory avoidance task. In the training phase, there was no difference in the transfer latency time among groups (Figure 8A). In the test phase, Aβ decreased (approximately 75%) the transfer latency time compared to the sham group (Figure 8B). NCTQ treatment significantly prevented this reduction (Figure 8B). Furthermore, NCTQ and TQ alone did not modify the time in step-down inhibitory avoidance (Figure 8B). These data demonstrated that only nanoencapsulated TQ protected against the impairment of non-spatial long-term aversive memory.

#### 3.4.2. Effects of Treatments or Aβ Peptide in Ex Vivo Analyses in Mice: Treatments Did Not Become Hepatotoxic at the Tested Dose in Mice

Table 4 shows the effect of treatments on the plasma biochemical markers, and there was no difference among groups in the ALT and AST activities. In this study, we verified that mice treated with TQ and NCTQ for 14 days at 1 mg/kg did not show changes in AST and ALT activities, suggesting that TQ and NCTQ were not hepatotoxic at the tested dose.

## 4. Discussion

This study aimed to develop and characterize physicochemical aspects of polymeric nanocapsules for TQ brain delivery, as well as the biological activity in a mouse model and the invertebrate *C. elegans*. The analytical method was developed and validated following the guidelines of the International Conference of Harmonization and the Brazilian Health Regulatory Agency [31,33]. NCTQ exhibited adequate physicochemical properties, including nanometer size and high encapsulation capacity. Treatment with NCTQ demonstrated low toxicity, reduced Aβ-induced paralysis in *C. elegans*, and activated an endoplasmic reticulum chaperone response. In the mouse model, Aβ treatment caused memory impairment, mitigated by NCTQ, improving working, long-term, and aversive memory. Furthermore, no significant changes in biochemical markers of hepatotoxicity in mice were observed.

Previously, our research group demonstrated that clozapine-loaded nanocapsules increased plasma exposure in rats treated with a single intravenous dose. The pharmacodynamic study also showed that nanoencapsulation improved the pharmacological effect in terms of antipsychotic potency as well as the action duration [22]. In this regard, Dimer et al. (2014) [43] demonstrated that nanoencapsulation of olanzapine increased brain exposure to the drug, corroborating the findings of our previous study. Different methods for obtaining nanocapsules are described in the literature, which are generally classified as in situ polymerization of monomers (alkyl cyanoacrylate) or the precipitation of pre-formed polymers (PCL, poly(lactic acid) (PLA), poly(lactic-co-glycolic acid) (PLGA), and methacrylate copolymers) [25,44]. The choice of preparation method, as well as the monomer or polymer to be employed, will depend on the study’s objectives, drug characteristics (if applicable), and desired production time [45]. In this study, the choice of nanoprecipitation method using PCL as the pre-polymer and Span 60^®^ as the surfactant was based on the previously observed good stability, reproducibility, and absence of toxicity [21,25].

Moreover, the nanoencapsulation of compounds has been associated with a decrease in compound toxicity. For instance, the co-encapsulation of an antimalarial agent (quinine) with curcumin demonstrated a protective effect on worms exposed to free quinine, as evidenced by survival and reproductive parameters [24]. Similar results were reported by Moraes et al., 2016 [46], with the nanoencapsulation of clozapine. Furthermore, clozapine-loaded nanocapsules demonstrated a lower incidence of oxidative damage in the brains of rats compared to the same dose of free drug. Additionally, blank nanocapsules were used as a control, showing no oxidative damage [23]. In this regard, the behavioral assessment of rats treated with blank nanocapsules with different coatings demonstrated no behavioral effects, especially those with anionic characteristics [25].

The nanometer size and polymer coating of polymeric nanocapsules positively improved physicochemical aspects and pharmacological efficacy [24,27]. These data corroborate other studies that developed PCL nanocapsules using polysorbate 80 as a stabilizer [22,24,27,47]. The obtained encapsulation efficiency values and drug content of approximately 100% demonstrated maximum drug entrapment capacity. These results are attributed to the oily core of the nanocapsules [22], which improves TQ solubilization. Considering the non-ionic nature of Span 60^®^, the negative zeta potential obtained was due to the ionization of the PCL carboxylic groups and agree with the previously reported results [25]. Furthermore, values close to −30 mV are considered adequate to ensure the balance between stability and cytotoxicity [48,49]. The zeta potential evaluation is important for the stability of the nanoparticle systems since particles with similar charges can repel each other, so with adequate charges, the samples are more resistant to flocculation, sedimentation, and aggregation processes [48]. The increase in drug exposure by nanoencapsulation emphasizes the importance of toxicological studies [22,50]. Alternative models provide preliminary information for initial safety screenings without using a significant number of mammals [24].

Notably, tellurium-containing organic compounds can cause toxic effects on animals, and some species may be more sensitive than others [16]. Given the absence of data on the effects of NCTQ in living organisms, we evaluated its toxicity in one non-mammalian model: *C. elegans* transgenic model of AD. Free TQ has been reported to lack toxicity at 1 µM [13], and our data demonstrated the safety of using TQ nanoformulations (Figure 4A–E). *C. elegans* has been proven to be a powerful complementary biological model during the preclinical drug development stage and studying senile diseases [51]. This is due to several mammalian orthologous genes, including numerous genes associated with human diseases [52]. When these human genes are not present in the worms, it is possible to generate transgenic animals that express these genes, presenting phenotypical characteristics that resemble human diseases. In this context, the strain CL2006 was bioengineered with a constitutive muscle-specific promoter, accumulating Aβ-immunoreactive deposits and intracellular amyloid (centrally involved in the AD pathogenesis), resulting in a progressive paralysis phenotype. Indeed, we observed that these worms presented high paralysis rates and that both TQ and NCTQ at 10 µM attenuated this phenotype (Figure 4E).

In a recent study, we found that TQ (1 µM) reversed oxidative damage and restored life expectancy reduced by the deleterious effects of paraquat by modulating the DAF-16/FOXO transcription factor [13]. The translocation of DAF-16 to the nucleus induced by TQ may activate antioxidant, pro-longevity, and proteostasis-promoting genes, which may help reduce the deleterious effects of Aβ. For instance, when DAF-16 is activated, it promotes a cellular survival response to the proteotoxicity induced by Aβ aggregation and polyglutamine expansion aggregation, acting in association with heat shock factor-1 (HSF-1) [53]. Therefore, based on the TQ mechanism previously evidenced, we hypothesized that this molecule could improve the phenotypical alterations caused by Aβ aggregation by activating molecular chaperones, also known as heat shock proteins (HSPs), which can degrade protein aggregates. Many chaperone families are involved in this process, such as the HSP70. Studies have demonstrated the role of HSP70 in preventing Aβ formation [54]. In this study, we demonstrated that TQ (1 and 10 µM) and NCTQ (10 µM) (Figure 5A–D) increased the activation of HSP-4 expression. 

In *C. elegans,* the HSP-4 is localized in the endoplasmic reticulum (ER), homologue to HSP70 family chaperones in humans, and is activated in response to stress conditions. Modulation in the DAF-16/FOXO through TQ treatment may be involved in the response of this chaperone [13]. However, mitochondrial HSP-6 did not increase by any of the treatments (Figure 6B–D). It is plausible that the treatments induced antioxidant enzymes due to DAF-16/FOXO activation and did not induce appropriate conditions for HSP-6 activation [13,55]. In addition, our data suggest that HSP-4 induction seems a compensatory response to contain Aβ formation and reduce the aggregates (Figure 5). It is important to emphasize that worms do not have as many cellular barriers as mammals, and their digestive system is quite different, which may explain the lack of differences between the free TQ and NCTQ in this study.

The pathogenesis of AD is closely associated with the presence of Aβ, which aggregates in the brain and forms senile plaques and intracellular neurofibrillary tangles, leading to neural loss and oxidative damage [1]. The most prominent Aβ fragments implicated in AD pathology are Aβ_40_ and Aβ_42_, which consist of 40 and 42 amino acids, respectively. For instance, Aβ_25–35_ is a widely investigated peptide fragment due to its ability to form aggregates and induce neurotoxic effects [56]. Therefore, an Alzheimer Aβ peptide-based model for mice was established and used to assess the effect of NCTQ. Indeed, this i.c.v. injection model with Aβ_25–35_ is commonly used as an AD model because it represents the physical and biological properties of diseases in mice (i.e., investigating exploratory behavior or cognitive function) related to spatial learning and memory [9,26].

The Y-maze is commonly used to evaluate spatial learning and working memory; this method is useful for assessing hippocampal damage [57]. A relevant finding was that the NCTQ treatment prevented the reduction of Aβ-induced alternations in the Y-maze task. At the same time, TQ had no effects (Figure 6B), proving that NCTQ could exert action in spatial and working memory impairment caused by Aβ. In this same sense, Ianiski et al. (2012) [26] reported the neuroprotective effects of nanoencapsulated meloxicam in mice with Aβ_25–35_-induced memory impairment, but not for free drugs, which was attributed to improved brain exposure to the drug. No locomotor alterations were observed since the treatments did not significantly affect the number of arm entries in the Y-maze test (Figure 6A).

The ORT is a valuable method for assessing LTM in mice, primarily involving hippocampal neuronal circuits in the consolidation of LTM [57]. The TQ and NCTQ prevented the reduction of the exploratory preference behavior associated with LTM (Figure 7B). These results further support previous studies indicating the potential of tellurium organic compounds as mnemonic enhancers [5,6,7,8]. Additionally, the step-down inhibitory avoidance task, used to assess aversive memory (also related to the hippocampus), demonstrated that the NCTQ protected the memory damage caused by the Aβ-peptide (Figure 8B). Notably, NCTQ treatment was more effective in protecting against impairment in aversive memory in animals pretreated with Aβ injection than those treated with TQ, thereby corroborating the data observed in the Y-maze test.

Corroborating these findings, our research group recently published a study demonstrating the enhanced effectiveness of anti-inflammatory agents (meloxicam and curcumin) when nanoencapsulated. This novel approach significantly improved the memory of mice exposed to Aβ_25–35_ compared to the effects observed with non-encapsulated drugs in the object recognition test [28].

Today, the drugs available for treating AD have several side effects that limit their use, such as hepatotoxicity [58,59]. Our results did not indicate any change in the plasma levels of these liver markers; despite being preliminary, these data, coupled with the absence of observed changes in locomotion or mortality, suggest the low toxicity of both TQ and NCTQ. Our research group recently conducted a study on a compound analogous to TQ, specifically 7-chloro-4-(phenylselanyl) quinoline, which similarly exhibited low toxicity in rodents [9].

This study also has other limitations that should be taken into consideration. Firstly, although the study suggests that nanoencapsulated TQ exerts neuroprotective effects by activating antioxidant pathways and the endoplasmic reticulum, the precise molecular mechanisms have not been fully elucidated. Furthermore, the study did not compare the effectiveness of TQ nanoencapsulation with existing treatments for AD. Moreover, the short-term evaluation in animal models does not account for the chronic nature of AD, warranting further investigation into long-term effects and potential disease-modifying properties. These limitations highlight the necessity for additional research to comprehensively assess the therapeutic potential of nanoencapsulated TQ to treat AD. Despite its limitations, this pioneering study on nanoencapsulated TQ provides crucial data, unlocking new possibilities for therapies and future research.

## 5. Conclusions

We developed NCTQ with satisfactory physicochemical characteristics. The preliminary analysis of *C. elegans* demonstrated its safety and efficacy in reducing Aβ peptide-induced paralysis rates in worms. The obtained data in the AD model in mice showed that NCTQ treatment did not present hepatotoxicity and attenuated the memory impairment caused by Aβ peptide. Notably, our data demonstrated that TQ nanoencapsulation improved the compound efficacy in rodents, suggesting that it is a promising candidate for further AD treatment assessments.

## Figures and Tables

**Figure 1 brainsci-13-00999-f001:**
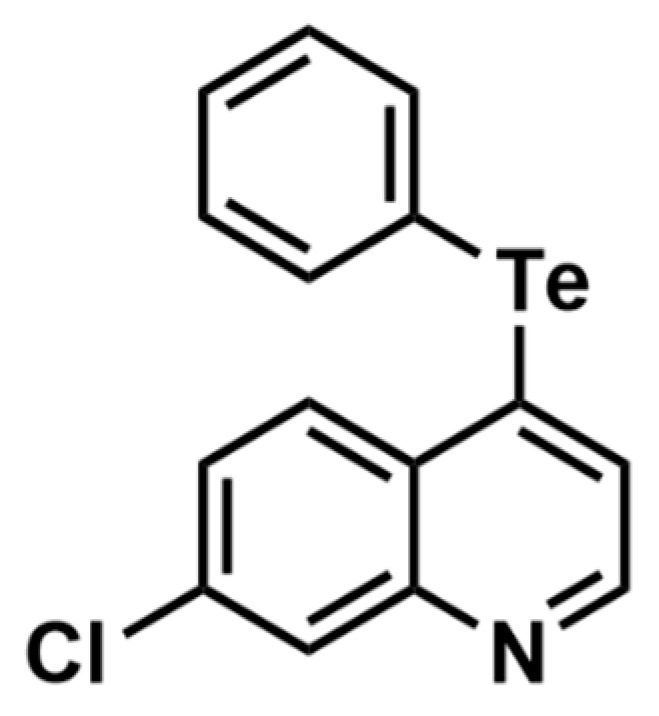
Molecular structure of TQ.

**Figure 2 brainsci-13-00999-f002:**
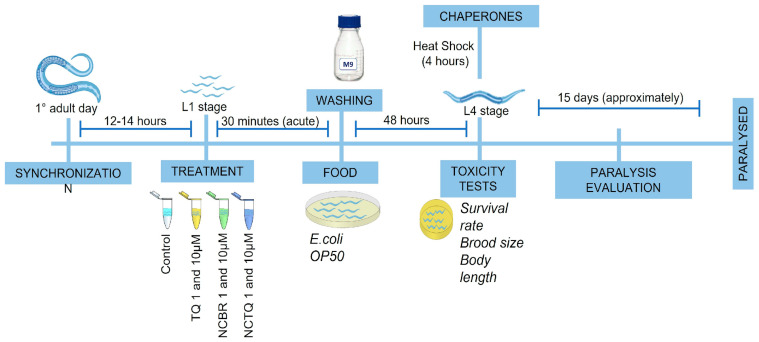
Scheme of the experimental protocol to assess the toxicity and efficacy of NCTQ in *C. elegans*. After 14 h of synchronization, worms in the first larval stage (L1) were acutely (30 min) exposed to TQ, NCTQ, or NCBR at 1 and 10 µM. Afterward, the worms were washed three times to remove the treatment. The animals were then placed on NGM plates seeded with *E. coli* OP50 as a food source until they reached the L4 larval stage. In this larval stage, we performed the safety evaluation through survival rate, brood size, and body length assays in the strain CL2006. In addition, in the L4 stage, worms of strains SJ4005 and SJ4100 were submitted to a heat shock for 4 h to posteriorly evaluation of chaperones HSP-4 and HSP-6 expression tagged with GFP. At the adult stage, we followed the Aβ aggregation through paralysis phenotype evaluation in the strain CL2006.

**Figure 3 brainsci-13-00999-f003:**
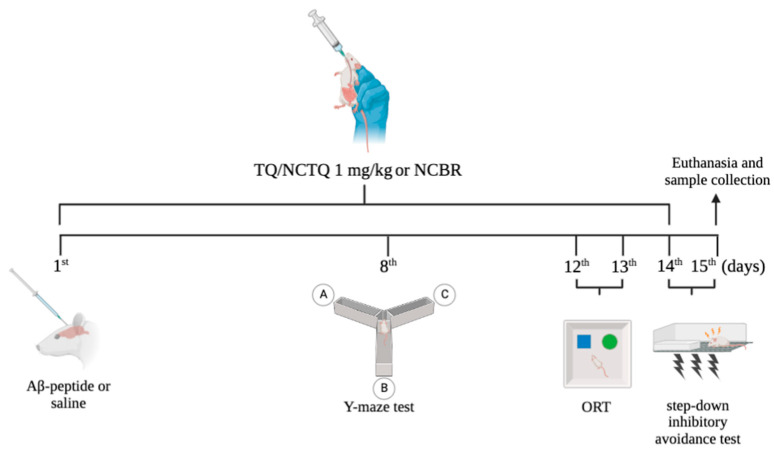
Scheme of the experimental protocol used to assess the general toxicity and efficacy of NCTQ in mice. On the first day, thirty minutes after intragastric treatments, mice received amyloid Aβ or vehicle (saline) by the intracerebroventricular route. I.g. treatments with TQ, NCTQ, and NCBR were performed every day until the fourteenth day of the experimental protocol. After four days of Aβ injection, the animals were submitted to the Y-maze, object recognition task, and step-down inhibitory avoidance tests. On the fifteenth day, blood was collected to evaluate general toxicity markers.

**Figure 4 brainsci-13-00999-f004:**
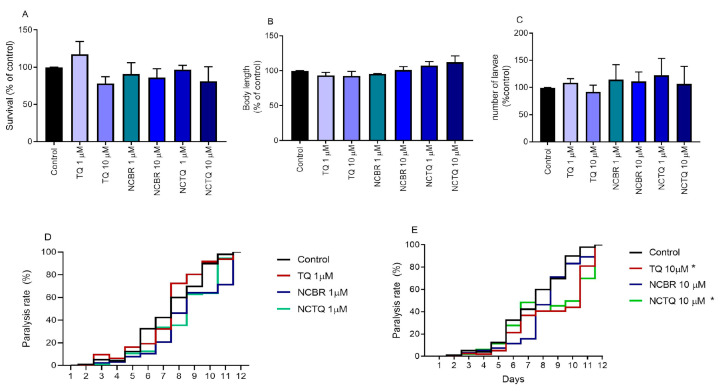
Toxicity endpoints of NCTQ in *C. elegans*. (**A**) Survival rate 48 h after the end of the treatment; (**B**) body length; (**C**) brood size. Data were analyzed by one-way ANOVA followed by Tukey’s post-hoc test; and paralysis rate caused by the expression of Aβ in *C. elegans* body–wall muscle cells and treated with free TQ or nanoformulation at (**D**) 1 µM; (**E**) 10 µM. Data were analyzed by RM one-way ANOVA followed by Dunnett’s post-hoc test. Asterisks indicate differences compared to the control group.

**Figure 5 brainsci-13-00999-f005:**
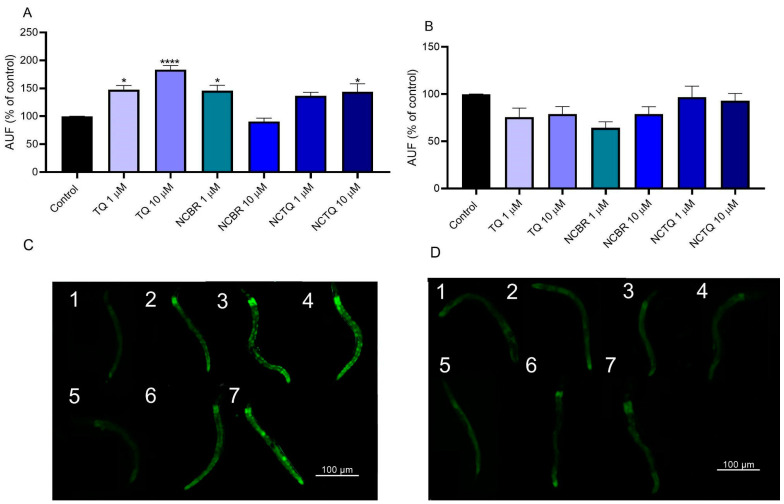
GFP intensity of reticular and mitochondrial chaperones in the strains (**A**) SJ4005 and (**B**) SJ4100. Data were analyzed by one-way ANOVA followed by Tukey’s post-hoc test; Asterisks indicate differences compared to the control group with * *p* < 0.05 and **** *p* < 0.0001. (**C**,**D**) are the representative images in strains S4005 and SJ4100, respectively: (1) Control, (2) TQ 1 µM, (3) TQ 10 µM, (4) NCBR 1 µM, (5) NCBR 10 µM, (6) NCTQ 1 µM, and (7) NCTQ 10 µM.

**Figure 6 brainsci-13-00999-f006:**
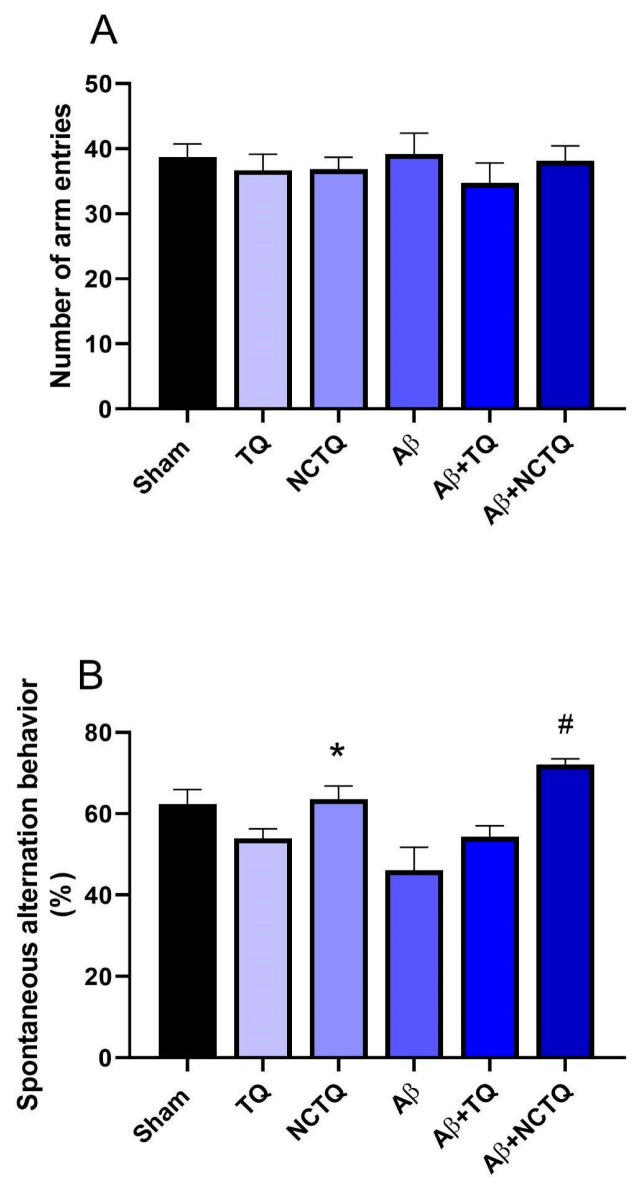
Effects of TQ, NCTQ, and/or Aβ on Y maze test in mice. (**A**) Number of arm entries; (**B**) spontaneous alternation behavior in the Y-maze task. Data are reported as mean ± standard error of the mean (SEM) of six to eight animals per group (one-way analysis of variance/Tukey’s test). (*) *p* < 0.05 as compared with the sham group. (#) *p* < 0.05 as compared with the Aβ group.

**Figure 7 brainsci-13-00999-f007:**
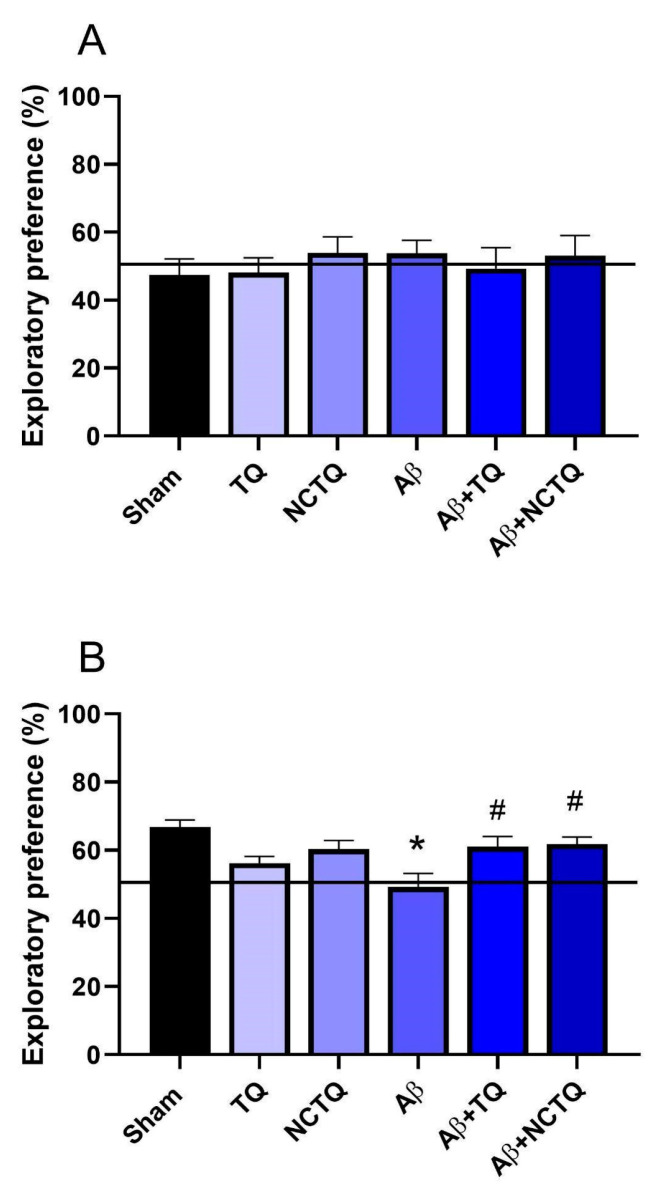
Effects of TQ, NCTQ, and/or Aβ on object recognition test in mice. (**A**) Time spent exploring one of the objects during the memory acquisition phase (training); (**B**) LTM (percentage of time spent exploring the novel object, a test carried out 24 h after training). Data are reported as mean ± standard error of the mean (SEM) of six to eight animals per group (one-way analysis of variance/Tukey’s test). (*) *p* < 0.05 as compared with the sham group. (#) *p* < 0.05 as compared with the Aβ group.

**Figure 8 brainsci-13-00999-f008:**
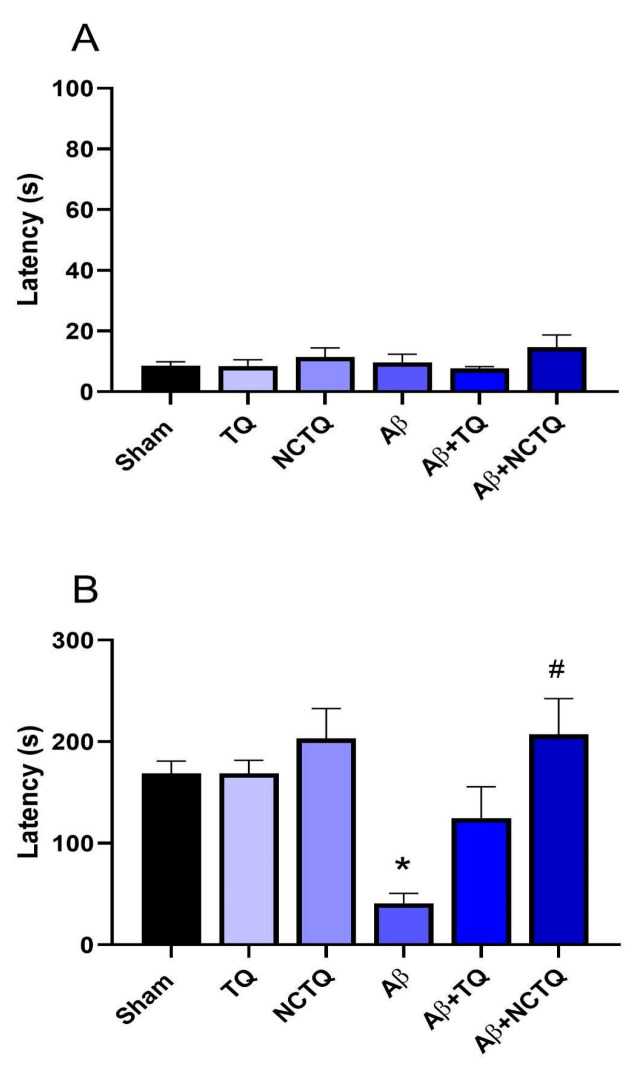
Effects of TQ, NCTQ, and/or Aβ on step-down inhibitory avoidance test in mice. (**A**) Latency (s) to fall from the platform in the step-down inhibitory avoidance in the training; (**B**) probe section. Data are reported as mean ± standard error of the mean (SEM) of six-eight animals per group (one-way analysis of variance/Tukey’s test). (*) *p* < 0.05 as compared with the Sham group. (#) *p* < 0.05 as compared with the Aβ group.

**Table 1 brainsci-13-00999-t001:** Physicochemical characterization of nanocapsules (*n* = 3).

	NCTQ	NCBR
D_[4,3]_ (nm)	240 ± 6	231 ± 6
Span	1.57 ± 0.04	1.58 ± 0.15
Zeta potential (mV)	−25.37 ± 1.52	−24.36 ± 5.54
pH	6.45 ± 0.07	6.29 ± 2.63
EE (%)	100	-
Drug content (%)	97.80 ± 0.37	-

**Table 2 brainsci-13-00999-t002:** Evaluation of repeatability/intermediate precision for nanoencapsulated TQ HPLC assay (*n* = 3).

TQ		
	Mean (%)	RSD (%)
Inter-day	99.77	1.15
Intra-day	99.87	1.27

The intra-day precision was carried out by preparing six samples of TQ containing 10 μg/mL. The inter-day precision was assessed on three different days (*n* = 18). The intra-day precision was assessed on the same day and with the same experimental conditions (*n* = 6).

**Table 3 brainsci-13-00999-t003:** Evaluation of robustness for nanoencapsulated TQ HPLC assay (*n* = 3).

Conditions	TQ					
	R*t* (min)	*T* (≤2.0)	*K* (≥2.0)	*N* (≥2000)	Mean (%)	RSD (%)
Proposed method *	8.20	1.7	4.5	8012	101.23	1
0.9 mL/min	8.90	1.7	2.4	7570	110.3	6.8
1.1 mL/min	7.20	1.7	2.4	7274	91.3	6.6
pH (6.8)	7.90	1.7	2.4	7338	100.5	0.33
pH (7.2)	7.80	1.7	2.4	7259	100.5	0.34

Mobile phase consisted of acetonitrile:water:triethylamine (90:10:0.3 *v*/*v*/*v*). The aqueous phase was adjusted to pH 7.0 with phosphoric acid; Nano Separation Technologies RP-18 column (4.6 mm × 150 mm); flow rate of 1.0 mL/min and TQ detection at 325 nm. R*t* = retention time; *T* = tail factor; *K* = capacity factor; *N* = the number of theoretical plates; RSD: relative standard deviation. * Mobile phase composed of water (0.03% of triethylamine (*v*/*v*)—pH 6.2) and acetonitrile (85:15 *v*/*v*); The flow rate was 1.0 mL/min.

**Table 4 brainsci-13-00999-t004:** Effect of TQ, NCTQ, and/or Aβ in the plasma biochemical markers in mice.

	Sham	TQ	NCTQ	Aβ	Aβ + TQ	Aβ + NCTQ
AST(U/L)	146.7 ± 5.2	168.6 ± 14.9	147.5 ± 10.2	157.5 ± 8.9	136.6 ± 9.1	174.9 ± 16.8
ALT(U/L)	35.2 ± 1.1	35.3 ± 5.7	45.1 ± 5.9	41.7 ± 5.6	39.7 ± 3.2	51.0 ± 4.1

Data are reported as mean ± standard error of the mean (SEM) of six to eight animals per group. Statistical analysis was performed using one-way analysis of variance/Tukey’s test. Aspartate (AST) and alanine (ALT) aminotrasferases.

## Data Availability

Data will be available if requested to the authors.

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
