# Peer review of "Development and In Vivo Assessment of 4-Phenyltellanyl-7-chloroquinoline-loaded Polymeric Nanocapsules in Alzheimer’s Disease Models"

_brainsci, 2023, doi:10.3390/brainsci13070999_

Round 1

Reviewer 1 Report

Comments and Suggestions for Authors

Review of a manuscript “Development and in vivo assessment of 4-phenyltellanyl-7- 2 chloroquinoline compound-loaded polymeric nanocapsules suspension in Alzheimer's disease models” by  Ana Cláudia Funguetto-Ribeiro and other coauthors submitted to the “Brain Sciences”.

Alzheimer’s disease is the most prevalent neurodegenerative disorder without efficient treatment affecting the main course of the disease. Therefore, new approaches for the therapy of Alzheimer’s disease are urgently needed. The authors consider a possibility of using nanoencapsulated drugs as a strategy for the targeted delivery of medications to the brain using as an example the delivery of nonionic surfactant polysorbate 80, curcumin, meloxicam, 4-Phenyltellanyl-7-chloroquinoline (TQ) and other substances. This is a very important area of biomedical research, and the results presented in the manuscript will be interesting for the readers of the journal.

The following corrections should be made:

Abstract

Lines 26-27: ”However, TQ use presents some deviations, such as low aqueous solubility and high toxicity…”

This is a clumsy sentence which can be corrected as follows: ”However, TQ possesses some drawbacks such as low aqueous solubility and high toxicity…”.

Lines 32-33: ”The treatment showed low toxicity, reduced Aβ peptide-induced paralysis and activated an endoplasmic reticulum chaperone in the C. elegans model, while Aβ-treatment in mice caused memory impairment, which NCTQ mitigated by improving working, long-term, and aversive memory”.

This is an awkward and too long sentence. It should be split into two easy to understand sentences. Furthermore, “Aβ-treatment in mice” may be understood as a cure by Aβ.

Line 35. “Additionally, no changes in biochemical markers were evidenced in mice”. This is incomplete sentence. No changes were evidenced as a result of what? Should be clarified.

Introduction

Line 40-41

The authors should begin Introduction by a brief characterization of AD, such as:” Alzheimer’s disease (AD) is a prevalent neurodegenerative disorder associated with other severe human diseases [Reference: ”Caveolin: a new link between diabetes and Alzheimer’s disease. Cell Mol Neurobiology, 2020 Jan 23. doi: 10.1007/s10571-020-00796-4]; so the search of new forms of medications is currently an urgent need.

Results

Figure 2

Some fronts of the text on Figure 2 are too small and their size should be increased.

Materials and Methods

Lines 192-193. ”The strains SJ4005 and SJ4100 present GFP tagging in the promoter of hsp-4 and hsp-

6, genes that codify the expression of chaperones localized in the endoplasmic reticulum and mitochondria, respectively”. The sense is unclear and should be clarified.

a)       “The strains SJ4005 and SJ4100 present GFP tagging in the promoter”.  Do the authors mean “The strains SJ4005 and SJ4100 contain GFP tags in the promoter?

b)      “genes that codify the expression” Do the authors mean : “genes encoding the expression”?

Line 211

2.5.1. Animals 210

“Three-month-old male Swiss albino mice weighing 25-35 g”

The weight of adult Swiss albino mice (male and female) is between 15–20 g.

Reference:

Tamta A, Chaudhary M, Sehgal R. A 28-days sub-acute toxicity study in swiss albino mice to evaluate toxicity profile of neurotol plus (mannitol and glycerol combination). Int J Biomed Sci. 2009 Dec;5(4):428-33. PMID: 23675168; PMCID: PMC3614800.

It is hard to believe that three-month-old male Swiss albino mice weigh 25-35 g.

Discussion

Line 537. “Even today, the drugs available for the treatment of AD have several side effects that limit their use, such as hepatotoxicity.” Reference should be added after the sentence

Author Response

Reviewer 1

            Review of a manuscript “Development and in vivo assessment of 4-phenyltellanyl-7- 2 chloroquinoline compound-loaded polymeric nanocapsules suspension in Alzheimer's disease models” by Ana Cláudia Funguetto-Ribeiro and other coauthors submitted to the “Brain Sciences”.

Alzheimer’s disease is the most prevalent neurodegenerative disorder without efficient treatment affecting the main course of the disease. Therefore, new approaches for the therapy of Alzheimer’s disease are urgently needed. The authors consider a possibility of using nanoencapsulated drugs as a strategy for the targeted delivery of medications to the brain using as an example the delivery of nonionic surfactant polysorbate 80, curcumin, meloxicam, 4-Phenyltellanyl-7-chloroquinoline (TQ) and other substances. This is a very important area of biomedical research, and the results presented in the manuscript will be interesting for the readers of the journal.

The following corrections should be made:

Abstract:

1) Lines 26-27: ”However, TQ use presents some deviations, such as low aqueous solubility and high toxicity…” This is a clumsy sentence which can be corrected as follows: ”However, TQ possesses some drawbacks such as low aqueous solubility and high toxicity…”.

Answer: The sentence has been corrected (Page 1, line 24-25).

-

2) Lines 32-33: ”The treatment showed low toxicity, reduced Aβ peptide-induced paralysis and activated an endoplasmic reticulum chaperone in the C. elegans model, while Aβ-treatment in mice caused memory impairment, which NCTQ mitigated by improving working, long-term, and aversive memory”. This is an awkward and too long sentence. It should be split into two easy to understand sentences. Furthermore, “Aβ-treatment in mice” may be understood as a cure by Aβ.

Answer: The sentence has been corrected (page 1, line 31).

 -

3) Line 35. “Additionally, no changes in biochemical markers were evidenced in mice”. This is incomplete sentence. No changes were evidenced as a result of what? Should be clarified.

Answer: The sentence has been rewritten (page 1, line 33-34).

 -

Introduction:

4) Line 40-41. The authors should begin Introduction by a brief characterization of AD, such as:” Alzheimer’s disease (AD) is a prevalent neurodegenerative disorder associated with other severe human diseases [Reference: ”Caveolin: a new link between diabetes and Alzheimer’s disease. Cell Mol Neurobiology, 2020 Jan 23. doi: 10.1007/s10571-020-00796-4]; so the search of new forms of medications is currently an urgent need.

Answer: Thank you for the suggestion. A new sentence containing additional information about AD has been inserted (page 1, lines 40-48).

-

Results:

5) Figure 2. Some fronts of the text on Figure 2 are too small and their size should be increased.

Answer: We appreciate your observation with Figure 2.  The authors made changes to the font size of the figure.

 -

Materials and Methods:

6) Lines 192-193. ”The strains SJ4005 and SJ4100 present GFP tagging in the promoter of hsp-4 and hsp-6, genes that codify the expression of chaperones localized in the endoplasmic reticulum and mitochondria, respectively”. The sense is unclear and should be clarified.

Answer: We appreciate your advice. The sentence is out of context. In this sentence of the article, the authors would like to highlight the role of green fluorescent protein (GFP) in marking the expression of two chaperones (HSP-4 and HSP-6). In addition, this is possible and specific to the target protein, when the GFP is inserted downstream of the promoter of these chaperones. Modifications were made in the text, with this  the sentence demonstrates more compatibility with this concept.

  1. a) “The strains SJ4005 and SJ4100 present GFP tagging in the promoter”. Do the authors mean “The strains SJ4005 and SJ4100 contain GFP tags in the promoter?

Answer: Sorry for the lack of clarity in the sentence. The authors mean ‘’the strains SJ4005 and SJ4100 present the expression of chaperones marked with GFP. Where the GFP is inserted downstream of the promoter of these chaperones’’

-

  1. b) “genes that codify the expression” Do the authors mean : “genes encoding the expression”?

Answer: Sorry for the lack of clarity in the sentence. Yes. The authors mean ''the genes encoding the chaperones HSP-4 and HSP-6 expression.''

 -

7) Line 211. 2.5.1. Animals 210

“Three-month-old male Swiss albino mice weighing 25-35 g”

The weight of adult Swiss albino mice (male and female) is between 15–20 g.

Reference:

Tamta A, Chaudhary M, Sehgal R. A 28-days sub-acute toxicity study in swiss albino mice to evaluate toxicity profile of neurotol plus (mannitol and glycerol combination). Int J Biomed Sci. 2009 Dec;5(4):428-33. PMID: 23675168; PMCID: PMC3614800.

It is hard to believe that three-month-old male Swiss albino mice weigh 25-35 g.

Answer: We appreciate the observation, but the average weight of our animals at 3 months of age ranges from 25 to 35 g. It should be noted that the age of these animals is an estimated average value, as the supply to research groups is based on demand and birth rates. Additionally, factors such as maintenance and composition of the feed may lead to variations in average weights among different research groups.

 References

Souza, A. C. G., Sari, M. H. M., Pinton, S., Luchese, C., Neto, J. S. S., & Nogueira, C. W. (2013). 2-Phenylethynyl-butyltellurium attenuates amyloid-β peptide(25-35)-induced learning and memory impairments in mice. Journal of Neuroscience Research, 91(6), 848–853. doi:10.1002/jnr.23211

Pinton, S., Souza, A. C., Sari, M. H. M., Ramalho, R. M., Rodrigues, C. M. P., & Nogueira, C. W. (2013). p,p′-Methoxyl-diphenyl diselenide protects against amyloid-β induced cytotoxicity in vitro and improves memory deficits in vivo. Behavioural Brain Research, 247, 241–247. doi:10.1016/j.bbr.2013.03.034

-

Discussion:

8)   Line 537. “Even today, the drugs available for the treatment of AD have several side effects that limit their use, such as hepatotoxicity.” Reference should be added after the sentence.

Answer: The reference has been added at the end of the sentence (page 16, line 568).

 -

Reviewer 2 Report

Comments and Suggestions for Authors

The manuscript “Development and in vivo assessment of 4-phenyltellanyl-7-chloroquinoline compound-loaded polymeric nanocapsules suspension in Alzheimer's disease models” requires improvement. Authors are strongly advised to revise and improve the English language and grammar throughout the manuscript.

Recommended title: Development and in vivo assessment of 4-phenyltellanyl-7-chloroquinoline-loaded polymeric nanocapsules in Alzheimer's disease models

4-phenyltellanyl-7-24 chloroquinoline, p should be in small letter, unless it appears as the first word in a sentence. Please revise throughout the manuscript.

In using abbreviations, once the abbreviation is introduced in full when it first appears, the abbreviated term should be used thereafter.

In using units, please use SI unit, eg. mg/mL. Use symbol degree for temperature °C. Please revise throughout the manuscript.

In writing scientific name for C. elegans and etc., make sure they are in italic.

Mistakes like nanocapsuled TQ or nanoencapsulated TQ? D[4,3] subscript? (line 287)

Introduction

In the last two sentences (first paragraph), the statements were not connected. Re-write and re-organise of these statements are strongly needed.

There are many nanoencapsulation techniques available. Please justify the use of polymeric nanoencapsulation in the present study.

Results

Values shown in 3.1 were not the same as in Table 1, eg. 240.00 ± 6.43 nm and 231.30 ± 6.11 nm. Please use approximately instead of the ~ in line 290.

Line 300: 08.2 minutes?

Provide higher resolutions Figures 4 and 5.

Discussion

Authors should discuss the specific nanoencapsulation technique used in the present study as compared to the literature.

Authors did not discuss the negative zeta potential obtained in the synthesised nanocapsules. Based on the values, they are less than the stable ±30 mV.

References

Reference style is not consistent, eg. some journal’s names were abbreviated while some were written in full.

Comments on the Quality of English Language

Proofreading is needed.

Author Response

Reviewer 2

            The manuscript “Development and in vivo assessment of 4-phenyltellanyl-7-chloroquinoline compound-loaded polymeric nanocapsules suspension in Alzheimer's disease models” requires improvement. Authors are strongly advised to revise and improve the English language and grammar throughout the manuscript.

1) Recommended title: Development and in vivo assessment of 4-phenyltellanyl-7-chloroquinoline-loaded polymeric nanocapsules in Alzheimer's disease models

Answer: Thank you for the suggestion. The title has been rewritten.

 -

2) 4-phenyltellanyl-7-24 chloroquinoline, p should be in small letter, unless it appears as the first word in a sentence. Please revise throughout the manuscript.

Answer: Thank you for the observation. The manuscript has been reviewed and corrected.

 -

3) In using abbreviations, once the abbreviation is introduced in full when it first appears, the abbreviated term should be used thereafter.

Answer: Thank you for the observation. The manuscript has been reviewed  and corrected.

 -

4) In using units, please use SI unit, eg. mg/mL. Use symbol degree for temperature °C. Please revise throughout the manuscript.

Answer: Thank you for the observation. The manuscript has been reviewed and corrected.

 -

5) In writing scientific name for C. elegans etc., make sure they are in italic.

Answer: Thank you for the observation. The manuscript has been reviewed and corrected.

6) Mistakes like nanocapsuled TQ or nanoencapsulated TQ? D[4,3] subscript? (line 287)

Answer: Thank you for the observation. The manuscript has been corrected.

-

Introduction:

7) In the last two sentences (first paragraph), the statements were not connected. Re-write and re-organise of these statements are strongly needed.

Answer: Thank you for the observation. The sentences were rewritten (page 2, line 53-58).

 -

8) There are many nanoencapsulation techniques available. Please justify the use of polymeric nanoencapsulation in the present study.

Answer: A brief justification for the choice of preparation technique has been added on page 2, lines 80-84. The modifications are highlighted in yellow.

 -

Results:

9) Values shown in 3.1 were not the same as in Table 1, eg. 240.00 ± 6.43 nm and 231.30 ± 6.11 nm.

Answer: Thank you for the observation. The manuscript has been corrected.

 -

10) Please use approximately instead of the ~ in line 290.

Answer: Thank you for the observation. The manuscript has been corrected.

 -

11) Line 300: 08.2 minutes?

Answer: The unit has been corrected.

 -

12) Provide higher resolutions Figures 4 and 5.

Answer. We appreciate your observation with Figures 4 and 5.  The authors made changes in the  figures.

 -

Discussion:

13) Authors should discuss the specific nanoencapsulation technique used in the present study as compared to the literature.

Answer: Thank you for the suggestion. Additional information about the different preparation techniques has been inserted on page 14, lines 453-461. The modification is highlighted in yellow.

 -

14) Authors did not discuss the negative zeta potential obtained in the synthesised nanocapsules. Based on the values, they are less than the stable ±30 mV.

Answer: Thank you for the suggestion. Additional information about the obtained zeta potential values has been inserted on page 14, lines 478-481. The modification is highlighted in yellow.

 -

References:

15) Reference style is not consistent, eg. some journal’s names were abbreviated while some were written in full.

Answer: We apologize for the error. The references have been carefully reviewed and corrected.

-

Round 2

Reviewer 1 Report

Comments and Suggestions for Authors I don't see that all my suggestions and critiques are met in the updated version. It still contains unclear text and some of my proposed additions were not added. For example, 1) line 212: "The strains SJ4005 and SJ4100 present GFP tagging the hsp-4 and hsp-6 expression, chaperones localized in the      endoplasmic reticulum and mitochondria, respectively" The sense is still unclear.

2) "they were exposed to a temperature up-shift at 37°C (heat shock) for 4 to induce chaperone expression [34]".

    What does it mean "4"?. The manuscript still contains unclear text and drawbacks as shown in these examples. 

Author Response

Reviewer #1 – Round 2

 I don't see that all my suggestions and critiques are met in the updated version. It still contains unclear text and some of my proposed additions were not added.

We apologize for the lack of clarity presented in the previous review. In fact, we realize that all the changes proposed to significantly be improving the quality of our manuscript. Please find below a point-by-point reply to the questions. All the changes made to the text are highlighted in yellow.

1) line 212: "The strains SJ4005 and SJ4100 present GFP tagging the hsp-4 and hsp-6 expression, chaperones localized in the endoplasmic reticulum and mitochondria, respectively" The sense is still unclear.

Answer: We regret the lack of clarity in the sentence in line 212. The sentence has been modified in the text for better understanding. "The strains SJ4005 and SJ4100 present a GFP reporter transgene controlled through of hsp-4 and hsp-6 promoter, the expression of these chaperones is localized in the endoplasmic reticulum and mitochondria, respectively"

We explain here briefly the meaning of this phrase:

SJ4005 and S4100 are two transgenic strains, in other words, these worms present genetic modifications in relation to the wild-type. These genetic modifications, including the GFP (Green Fluorescent Protein) reporter transgene insertion. The GFP is called a transgene, because it was isolated from other organism (Aequorea victoria) and inserted in the C.elegans material genetic. And the reporter referred to the ability of the GFP, when introduced into target cells, produced a green fluorescent tagging the protein of interest.

In the case of strains SJ4005 and SJ4100, the GFP reporter transgene presents the expression controlled by the same promoters involved in the hsp-4 and hsp-6 expression. (Page 5, line 212-214)

2) "they were exposed to a temperature up-shift at 37 °C (heat shock) for 4 to induce chaperone expression [34]". What does it mean "4”? The manuscript still contains unclear text and drawbacks as shown in these examples. 

Answer: We regret the lack of attention in writing this methodology. The sentence has been rewritten, and 4 refers to the period of 4 hours. With relation to the unclear text and drawbacks, the authors performed a review according to the reviewers' suggestions. I hope to meet your expectations. (Page 5, line 215)

3) Are the Methods adequately described? Must be improved

Answer: The methods are described, performed, and standardized by our research group in accordance with previous studies. It is possible to verify in these references Pereira et al. (2022) and Canedo-Reis et al. 2022.  Necessary modifications made to better understanding the methods are highlighted in the manuscript (2.4.2, 2.4.3 and 2.4.4 section). Furthermore, additional information regarding the Y-maze task was included (section 2.5.3.1) to provide a better understanding of the test, resulting in minimal modifications to Figure 3.

References

Canedo-Reis, N. A. P., de Oliveira Pereira, F. S., Ávila, D. S., Guerra, C. C., Flores da Silva, L., Junges, C. H., ... & Bergold, A. M. (2022). Grape juice reduces the effects of amyloid β aggregation phenotype and extends the longevity in Caenorhabditis elegans. Nutritional Neuroscience, 1-12.

de Oliveira Pereira, F. S., Barbosa, F. A. R., Canto, R. F. S., Lucchese, C., Pinton, S., Braga, A. L., ... & Avila, D. S. (2022). Dihydropyrimidinone-derived selenoesters efficacy and safety in an in vivo model of Aβ aggregation. NeuroToxicology, 88, 14-24.

-

Reviewer #1 – Round 1

Abstract:

1) Lines 26-27: ”However, TQ use presents some deviations, such as low aqueous solubility and high toxicity…” This is a clumsy sentence which can be corrected as follows: ”However, TQ possesses some drawbacks such as low aqueous solubility and high toxicity…”.

Answer: The sentence has been corrected (Page 1, line 24-25).

-

2) Lines 32-33: ”The treatment showed low toxicity, reduced Aβ peptide-induced paralysis and activated an endoplasmic reticulum chaperone in the C. elegans model, while Aβ-treatment in mice caused memory impairment, which NCTQ mitigated by improving working, long-term, and aversive memory”. This is an awkward and too long sentence. It should be split into two easy to understand sentences. Furthermore, “Aβ-treatment in mice” may be understood as a cure by Aβ.

Answer: The sentence has been corrected (page 1, line 31).

 -

3) Line 35. “Additionally, no changes in biochemical markers were evidenced in mice”. This is incomplete sentence. No changes were evidenced as a result of what? Should be clarified.

Answer: The sentence has been rewritten (page 1, line 33-34).

 -

Introduction:

4) Line 40-41. The authors should begin Introduction by a brief characterization of AD, such as:” Alzheimer’s disease (AD) is a prevalent neurodegenerative disorder associated with other severe human diseases [Reference: ”Caveolin: a new link between diabetes and Alzheimer’s disease. Cell Mol Neurobiology, 2020 Jan 23. doi: 10.1007/s10571-020-00796-4]; so the search of new forms of medications is currently an urgent need.

Answer: Thank you for the suggestion. A new sentence containing additional information about AD has been inserted (page 1, lines 40-49).

-

Results:

5) Figure 2. Some fronts of the text on Figure 2 are too small and their size should be increased.

Answer: We appreciate your observation with Figure 2.  The authors made changes to the font size of the figure.

 -

Materials and Methods:

6) Lines 192-193. ”The strains SJ4005 and SJ4100 present GFP tagging in the promoter of hsp-4 and hsp-6, genes that codify the expression of chaperones localized in the endoplasmic reticulum and mitochondria, respectively”. The sense is unclear and should be clarified.

Answer: We appreciate your advice. The sentence is out of context. In this sentence of the article, the authors would like to highlight the role of green fluorescent protein (GFP) in marking the expression of two chaperones (HSP-4 and HSP-6). In addition, this is possible and specific to the target protein, when the GFP is inserted downstream of the promoter of these chaperones. Modifications were made in the text, with this  the sentence demonstrates more compatibility with this concept.

  1. a) “The strains SJ4005 and SJ4100 present GFP tagging in the promoter”. Do the authors mean “The strains SJ4005 and SJ4100 contain GFP tags in the promoter?

Answer: Sorry for the lack of clarity in the sentence. The authors mean ‘’the strains SJ4005 and SJ4100 present the expression of chaperones marked with GFP. Where the GFP is inserted downstream of the promoter of these chaperones’’

-

  1. b) “genes that codify the expression” Do the authors mean : “genes encoding the expression”?

Answer: Sorry for the lack of clarity in the sentence. Yes. The authors mean ''the genes encoding the chaperones HSP-4 and HSP-6 expression.''

 -

7) Line 211. 2.5.1. Animals 210

“Three-month-old male Swiss albino mice weighing 25-35 g”

The weight of adult Swiss albino mice (male and female) is between 15–20 g.

Reference:

Tamta A, Chaudhary M, Sehgal R. A 28-days sub-acute toxicity study in swiss albino mice to evaluate toxicity profile of neurotol plus (mannitol and glycerol combination). Int J Biomed Sci. 2009 Dec;5(4):428-33. PMID: 23675168; PMCID: PMC3614800.

It is hard to believe that three-month-old male Swiss albino mice weigh 25-35 g.

Answer: We appreciate the observation, but the average weight of our animals at 3 months of age ranges from 25 to 35 g. It should be noted that the age of these animals is an estimated average value, as the supply to research groups is based on demand and birth rates. Additionally, factors such as maintenance and composition of the feed may lead to variations in average weights among different research groups.

 References

Souza, A. C. G., Sari, M. H. M., Pinton, S., Luchese, C., Neto, J. S. S., & Nogueira, C. W. (2013). 2-Phenylethynyl-butyltellurium attenuates amyloid-β peptide(25-35)-induced learning and memory impairments in mice. Journal of Neuroscience Research, 91(6), 848–853. doi:10.1002/jnr.23211

Pinton, S., Souza, A. C., Sari, M. H. M., Ramalho, R. M., Rodrigues, C. M. P., & Nogueira, C. W. (2013). p,p′-Methoxyl-diphenyl diselenide protects against amyloid-β induced cytotoxicity in vitro and improves memory deficits in vivo. Behavioural Brain Research, 247, 241–247. doi:10.1016/j.bbr.2013.03.034

-

Discussion:

8)   Line 537. “Even today, the drugs available for the treatment of AD have several side effects that limit their use, such as hepatotoxicity.” Reference should be added after the sentence.

Answer: The reference has been added at the end of the sentence (page 16, line 580).